# Application of the Avrami Equation to the Dilatometric Analysis of ADI Austempering Kinetics

**DOI:** 10.3390/ma18215039

**Published:** 2025-11-05

**Authors:** Tomasz Wiktor, Andriy Burbelko, Artur Zaczyński

**Affiliations:** 1Faculty of Foundry Engineering, AGH University of Krakow, 30 Mickiewicza Avenue, 30-059 Krakow, Poland; 2Odlewnie Polskie Spółka Akcyjna, Inżynier Władysława Rogowskiego 22 Street, 27-200 Starachowice, Poland; artur.zaczynski@odlewniepolskie.pl

**Keywords:** ADI, austempering kinetics, dilatometry, KJMA theory, Avrami transformation, screening effect

## Abstract

The method and results of evaluating the kinetics of austenite isothermal decomposition in austempered ductile iron (ADI) samples are presented based on the dimensional changes in austenitized and isothermally hardened cast iron samples. Experimental measurements were carried out on samples intended for the production of ADI castings under industrial conditions of ODLEWNIE POLSKIE S.A. A partial solution of the Kolmogorov–Johnson–Mehl–Avrami statistical theory of phase transformations as proposed by Avrami was applied to analyze the experimental results of dilatometric measurements. It is shown that Avrami diagrams can be used to evaluate changes in the kinetics of phase transformations occurring in ADI samples during the first stage of isothermal austenite decomposition. The application of the proposed method has made it possible to identify three steps of ausferrite growth during the first stage, with two statistically significant slowdowns. Using quantitative metallography methods, it is demonstrated that the slowdown in the rate of austenite decomposition during the transition from the first to the second step is related to the development of the microstructure of the metallic matrix of cast iron.

## 1. Introduction

Austempered ductile iron (ADI) is an advanced material for making heat-treated castings. The properties of this material combine high strength, ductility and impact strength with good wear resistance and machinability. These properties can be achieved after appropriate heat treatment, which provides the optimal microstructure for a given wall thickness and chemical composition. The goal of such heat treatment is to obtain a metal matrix microstructure of ADI called ausferrite. This structure is composed of metastable carbon-supersaturated austenite and the precipitates of ausferritic ferrite.

Heat treatment involves austenitizing the cast iron and saturating the resulting austenite with carbon. The castings are then rapidly cooled to isothermal quenching temperature in order to suppress the pearlite formation. The temperature selected is at a level that does not cause martensitic transformation.

During isothermal quenching, ferrite, whose grains have the shape of jagged plates, is first released from carbon-saturated austenite [1]. The grains of ausferritic ferrite are free of precipitates of other phases. In earlier publications, this ferrite was referred to as “bainite” [2,3] or termed as “bainitic” [4,5] or “acicular” ferrite [6]. Appropriate names for lamellar ferrite grains were proposed in the work of [3] as “wheat-ear-like” or “resembling the shape of an ear of wheat” and in [7] as “sheaf”. The growth of lamellar ferrite α causes the redistribution of carbon between the grains of this phase with a relatively low solid solubility limit of this element and the austenitic matrix, causing supersaturation of austenite. This constitutes the first stage of the isothermal decomposition of austenite γ described by the following expression:(1)$$\gamma \to \alpha +\gamma^{\prime },$$
where γ′ is austenite supersaturated by carbon.

This procedure is usually carried out from 250 to 400 °C. Austenite with a higher carbon content has a lower temperature for the onset of martensitic transformation. At ambient temperature, high-carbon austenite can also remain in a metastable state indefinitely.

However, an excessive extension of the isothermal quenching time beyond the limit of the so-called process window will result in a decrease in the mechanical properties. The reason for this is the destruction of high-carbon austenite, during which carbide particles begin to separate from it, which usually results in a decrease in density and an increase in alloy volume:(2)$$\gamma^{\prime }\to \alpha +\mathrm{carbides}.$$

This transformation constitutes the second stage of isothermal austenite decomposition.

The issue of quickly determining the maximum length of the so-called process window has been the focus of many researchers for years. Various methods are used to determine the limits of the process window. As a rule, for the analyzed temperature and austenitizing time, the series of samples is cooled after the high-temperature treatment to the austenite decomposition temperature. Subsequent samples from this series after various austempering periods are rapidly cooled to ambient temperature for microstructure and property studies [8].

Most often, the testing of samples obtained in this way is performed by means of X-ray diffraction combined with optical or electron microscopy and, if necessary, hardness measurements [5,9,10,11,12,13,14]. The combination of X-ray diffraction with metallography has also been employed in [15] for kinetics analysis of thermo-mechanical processing of ADI. A comparison of the magnetic method, XRD and quantitative metallography is presented in the publication [1].

The research methods outlined above provide in-depth information about the nature and kinetics of phase transformations. Unfortunately, due to time considerations, they cannot be used for a reasonably quick determination of the limits of the process window, which may be necessary under industrial conditions when producing small batches of ADI castings.

## 2. State of the Art

### 2.1. Dilatometric Studies of Austenite Decomposition in ADI

To directly assess the effect of process conditions (temperature and quenching time) on the development of phase transformations during the formation of the metal matrix microstructure, ADI, Santos et al. [16] used a quenching dilatometer. The kinetics of austenite decomposition during cooling and isothermal transformation have been described using a graphical relationship between the change in relative elongation and the logarithm of time. Based on these relationships, the transformation time was determined by four temperature levels of isothermal processing. The logarithm of the transformation time was found to be linearly related to the temperature of the isothermal treatment.

The authors of the study introduced a correction for the transition process, i.e., the cooling time in the range from the austenitizing temperature to the isothermal transformation temperature. This correction considers the apparent linear coefficient of thermal expansion. The diagrams obtained from these relations have the shape of a smooth sigmoidal function. However, the article does not provide information on how to select the moments of time corresponding to the beginning and end of austenite decomposition.

Optimization of the two-step ADI heat treatment was presented by Gazda et al. [17]. A quantitative dilatometric analysis combined with the calorimetric (DSC) method was used for the analysis of the ausferrite decomposition kinetics. These tools allow excellent mechanical properties of ADI after the temperature step-down heat treatment. The relations between relative elongation and temperature obtained in the experiments were reconvoluted by PeakFit software. The results of the dilatometric tests are presented in terms of temperature dependence.

The dilatometry studies for the developed ADI alloys presented by Ahmed et al. [18] showed that the time required for the completion of the ausferrite formation in the alloys solidify under the conditions of the ultrasonic melt treatment was four times shorter than that required for statically solidified SG irons.

In a publication by Lachart et al. [19], it was proven that the dilatometric method can predict microstructure variation in heavy section parts with small samples only. Dilatometric tests were performed on small samples for cooling rates corresponding to the conditions found at different locations in the test casting with varying wall thicknesses. Based on the results obtained, it is possible to predict the risk of perlitic structure in areas with low rates of temperature change during quenching.

Górny et al. [20] used the dilatometric method in combination with other methods of structure studies to evaluate the effect of the temperature of the austenitizing procedure on the course and rate of phase transformations occurring in the isothermal hardening process. Two chemical compositions of ADI and four levels of austenitization temperature were analyzed. The influence of the above parameters was presented in terms of the transformation rate versus time relationship, the maximum solidification rate and the time at which the maximum rate was reached.

### 2.2. Mathematical Models of Austenite Decomposition Kinetics in ADI

The fundamentals of the general statistical theory of phase transitions involving nucleation and growth of new phase grains were published by Kolmogorov [21]. Based on an analysis of the probability that a random point remains in the parent phase, this article proposes a general solution for determining the volume fraction of transformation products, *f*. The solution relates to the uniform in space random nucleation of grains with varying speed. The grains were assumed to have a convex shape, and it was further assumed that grains can have anisotropic geometry, provided they have the same spatial orientation. The migration speed of the interfacial “solid–liquid” boundary can be variable. In addition to the general solution, two partial solutions for three-dimensional growth with constant velocity are presented in [21]. Both solutions were obtained for a constant interfacial migration velocity. One concerns a constant nucleation rate. The other is for so-called instantaneous nucleation (constant grain number). These solutions take the form of(3)$$f=1-\exp\left(-k\cdot t^{n}\right),$$
where *t* is time and the *k* and *n* parameters depend on the shape of the grains, the migration rate of the interfacial boundary and the nucleation rate.

Equations of this type, proposed in the 1930s and 1940s and also in publications [22,23,24,25], are now assumed to be called Kolmogorov–Johanson–Mehl–Avrami (KJMA) equations. The Kolmogorov model can be used to analyze both crystallization processes and solid-state phase transformations. Moreover, it is applicable not only in material science but also in the biology of ecological landscapes [26]. This model describes the kinetics of the processes involving the emergence of growth centers and the subsequent increase in the size of the areas surrounding these centers.

The first mathematical model of the kinetics of the isothermal transformation of austenite in the temperature range of the bainitic transformation was obtained by Austin and Rickett (AR) based on the approximation of experimental data [27]:(4)$$\frac{f}{f_{\max}}=1-\frac{1}{1+k\cdot t^{n}},$$
where *f* is the volume share of transformation products, *f*_max_ is the maximum share that can be achieved at the analyzed temperature, and *k* and *n* are the temperature-dependent constants.

Avrami [24] represents this equation in the following form:(5)$$\frac{f\left(t\right)}{1-f\left(t\right)}=B\cdot t^{n}.$$

Starink [28] compared the accuracy of mapping the kinetics of a diffusion-controlled precipitation reaction using the KJMA and AR equations under isothermal conditions for six different alloys. It is shown that, if only one type of precipitation process occurs simultaneously, the AR equation immediately provided a better fit to the experimental data sets. The paper [28] noted that the reason for the deviation of the prediction using the KJMA equation from the experimental data is the failure to meet the assumptions made in [21].

The article by Starink [29] presents a unique kinetic model:(6)$$f=1-{\left[\frac{{\left(k\cdot t\right)}^{n}}{\eta_{i}}+1\right]}^{-\eta_{i}},$$
which, in addition to the parameters *k* and *n* described above, also includes the impingement parameter η*_i_*. The purpose of this parameter is to account for the influence of the shielding effect on the growth of ferrite grains shaped like wheat ears.

An exponential relation between the transformed volume fraction *f* and process time *t* was proposed in [20]:(7)$$f=1-\exp\left(-\frac{t}{k}\right),$$
where *k* is a curve shape constantly valid for a given transformation condition.

### 2.3. Description of Model Used: KJMA Equation and Avrami Diagram

The KJMA equation in the form of Equation (3) was used for the numerical simulation of in situ heat treatment of austempering ductile iron in [30] to calculate the amounts of pearlite and of ausferrite. The parameters of this equation have been determined at each temperature from the “start” and “end” of the isothermal transformation curves on the TTT diagram.

The numerical simulation of isothermal austempering heat treatment of a ductile cast iron presented by Boccardo et al. [31] used the KJMA equation with a constant exponent. The model with an exponent of *n* = 2.1 correctly predicts the course of austenite decomposition compared with the experimental results course obtained in the described experiment.

Pereira et al. [32] used the KJMA equation for the kinetics of ferrite growth during austempering for three different austenitization routes: two-step, conventional and rapid austenitization. For each of the analyzed methods, eight samples were used to measure the volume fraction of ferrite. These samples were rapidly cooled from the austempering temperature after holding for 15 min to 6 h. The exponent *n* in the range 1.5…1.7 was estimated using nonlinear regression—Origin^®^ 2019 software on the base of collected data.

Suchocki et al. [33] underscore that a significant improvement in the performance of the JMAK model was achieved when the kinetic constants were determined using the least squares method. In this work, the Scilab software was used to process the data from the dilatometric measurements. The obtained exponent *n* values are placed in the range from 0.948 (for austenite decomposition in the temperature 500 °C) to 2.1082 (for 270 °C).

To better understand the physical meaning of the exponent *n* in Equation (4), let us look at the general solution proposed by Kolmogorov [21], which can be expressed as follows:(8)$$f=1-\exp\left[-\frac{4\pi }{3}c^{3}{\int_{t_{0}}^{t}\alpha \left(t^{\prime }\right)\left(\int_{t^{\prime }}^{t}k\left(\tau \right)d\tau \right)}^{N_{\dim}}dt^{\prime }\right],$$
where *c* is the grain shape constant, α(*t*′) is the time-varying rate of nucleation, *u*(τ) is the time-dependent migration speed of the boundary of the growing grain, and *n* is the dimensionality of the growth.

Avrami clarified the physical meaning of the argument of the exponential function in the Kolmogorov Equation (8) by introducing the concept of extended volume *V*_ex_, “which is the total transformed volume if we neglect overlapping of growing grains” [23]. For a constant number of spherical grains (*N*_dim_ = 3) and their growth at a constant velocity *u*, the extended volume is calculated as follows:(9)$$V_{\mathrm{ex}}=N_{g}\frac{4\pi \cdot u^{3}t^{3}}{3},$$
where *N*_g_ is the grain number per unit volume, and(10)$$f=1-\exp\left(N_{g}\frac{4\pi u^{3}t^{3}}{3}\right),$$

Avrami [27] noted that, for constants *n* and *k*, applying a double logarithmic transformation of Equation (3) leads to the following linear equation relationship:(11)$$\ln\left[-\ln\left(1-f\right)\right]=\ln k+n\ln t.$$

The application of Equation (11) for the analysis of the kinetics of austempering reactions in ductile irons for four types of cast iron is described in [33]. The experimental results obtained using the dilatometric method were analyzed. In the publication [34], the parameters of the Avrami equation for the analysis of the kinetics of the isothermal decomposition of austenite were determined based on nine measurements of changes in the hardness of the samples. These samples were subjected to isothermal treatment interrupted after a period of 1 min to 180 min.

Comprehensive studies of phase transition kinetics in ADI using neutron diffraction, dilatation measurements, atom probe tomography and metallographic methods are presented in [35]. It has been shown that this equation does not comprehensively describe stage I and stage II of isothermal transformation, but it was shown that dilatometry is a well-established method that can determine the time for maximum C enrichment in ADI during austempering.

An evaluation of different models of the transformation kinetics of isothermal austempering transformation was presented in [36]. To this purpose, a series of dilatometric tests were carried out on ADI.

The physical meaning of parameter *k* in this equation can be derived from a compilation of Equations (10) and (11). For spherical growth with a constant velocity, the equation is as follows:(12)$$k=N_{g}4/3\pi u^{3}.$$

Starink [29] pointed out the dependence of the exponent *n* on the nucleation rate and changes in growth velocity of transformation products as follows:(13)$$n=N_{\dim}\cdot g+B.$$

For continuous nucleation with a constant rate, *B* = 1, whereas for instantaneous nucleation, *B* = 0. If the boundary migrates at a constant rate, the value of *g* is 1. If the linear growth rate increases (e.g., due to increased undercooling), *g* > 1.

If the growth rate changes over time according to a power function $u=k_{u}t^{n^{\prime }}$, the extended volume will be equal to(14)$$V_{\mathrm{ex}}=N_{g}\frac{4\pi k_{u}^{3}}{3{\left(n^{\prime }+1\right)}^{3}}t^{3\left(n^{\prime }+1\right)}.$$

In this case,(15)$$k=N_{g}\frac{4\pi k_{u}^{3}}{3{\left(n^{\prime }+1\right)}^{3}}.$$

For 3D growth limited by diffusion, theoretically, *n*′ = −1/2, which means that the value of the exponent *n* = 1½.

The deviation of the Avrami diagram, i.e., the graph of the left side of Equation (8) versus the natural logarithm of time, from a straight line may indicate a change in the transformation mechanism. The use of such a transformation provides the opportunity to detect moments of change in the kinetics of phase transformation processes based on an analysis of the graphical representation of the experimental results.

As can be seen in Equations (8) and (13), changes in the exponent *n* during the transformation can be associated with variations in the nucleation rate, the migration velocity of the boundary and/or the dimensionality of growth. During crystallization and solid-state phase transformations, the nucleation period can be shorter than the time required for complete transformation. If the speed and dimensionality of growth remain unchanged, the moment of nucleation termination can be seen in the Avrami diagrams. An indicator of the end of nucleation will be a decrease in the angular coefficient of the diagram.

These factors may not be accounted for in the general Avrami equation for transformation with very short incubation periods compared to the time required to complete the reaction. An illustrative example of this is provided in [37].

A summary of example values of the exponent *n* for different transformation mechanisms is provided in [38]. If both the migration speed of the boundary and the nucleation rate do not increase, the exponent *n* does not exceed four. The minimum value of *n* is equal to ½ and applies to one-dimensional, diffusion-controlled growth (e.g., for large plates that no longer grow in the longitudinal directions due to full impingements of the sidewalls).

Another reason for reducing the slope of the Avrami diagram may be a reduction in the dimensionality of growth. Changes of this type are caused by the blocking effect (also referred to as “shielding” or “screening”) [29,39,40]. A quantitative assessment of the correction to Equation (3) in the case of the anisotropic shape of the grains and their random spatial orientation is described in [41].

For needle-shaped grains, three-dimensional growth will be reduced to two-dimensional after full impingements of the rapidly growing end walls. A similar situation after impingements of the sidewalls applies to the growth of thin plates. If the growth rate of plate grains varies depending on the crystallographic orientation of the growing surface, the growth dimension will decrease from the initial three to two and then to one, as the faster end walls will be blocked by the slower growing surfaces of adjacent grains.

One more factor that can change the linearity of the Avrami diagram is a change in the mechanism of the phase transformation. In the case of ADI heat treatment using the austempering method, such a change may occur because of the transition from the first to second stage of the process and possibly during the first stage.

The aim of the research described in this article is to examine the possibility of analyzing changes in the kinetics of the first stage of austenite decomposition in the isothermal austempering process.

An excessive prolongation of low-temperature treatment is risky because it may lead to the onset of the second stage and deterioration of the properties of cast iron. This means that increasing the duration of this treatment is risky. On the other hand, it is not very effective due to the constantly decreasing rate of austenite decomposition. Compared to methods involving the preparation and testing of a series of samples, dilatometric analysis is continuous from a practical point of view. Therefore, the results of such an analysis can serve as a basis for a justified and more precise reduction in the time required for this procedure. It should also be noted that switching from a series of quenches to “online” dilatometry not only increases the accuracy of the time assessment but also speeds up the analysis and reduces the cost of testing.

## 3. Apparatus, Material, Heat Treatment Mode and Measurements

The following section provides a detailed description of the apparatus, the materials used and the heat treatment mode employed.

### 3.1. Apparatus

The dilatometric analysis was carried out in the laboratory installation developed at the Faculty of Foundry Engineering, AGH University, Krakow, Poland, for isothermal quenching of cast iron, shown in Figure 1. A sample with a diameter of 4 mm and a length of 30 mm was placed in a stationary vertical quartz tube (1), closed at the bottom. The top end of the sample’s quartz tube was connected to the base of a digital indicator (5). Changes in sample length were transmitted to the measuring device through a tubular stylus, also made of quartz. Throughout the experiment, technical argon was fed into the specimen workspace to prevent oxidation and decarburization of the surface.

Two furnaces were used for the heat treatment: a high-temperature vertical resistance tube furnace for the austenitization treatment (2) and a salt bath for isothermal quenching (3) filled with a mixture of melted NaNO_2_–KNO_3_. Both furnaces were placed on a platform to ensure that they could be moved vertically and horizontally (4). During heating and austenitizing, the tube with the sample was in the working space of the furnace (2). After the austenitizing period, the furnaces were moved in a downward direction.

Due to the small volume of the salt bath, the quartz tube with the sample was pre-cooled in a container of water to prevent the salt temperature from rising excessively during quenching. The furnaces were then moved horizontally and lifted to place the sample tube in the salt bath.

The temperature of the specimen was measured using a Type S thermocouple, the wires of which were contact welded to the top face of the specimen. The measurement results were recorded with an Agilent 34970A Data Acquisition/Switch Unit (Keysight, 1400 Fountaingrove Parkway, Santa Rosa, CA, USA) with a sampling time of 4.04 s. All changes in probe length at the time of heating, cooling and austempering treatment were measured using a digital indicator SYLVAC S_Dial Work Nano, model 805-5306 with 0.1 μm resolution (Sylvac, Avenue des Sciences 19, 1400 Yverdon, Switzerland). Measuring force: standard—0.65–0.9 N, low—0.4–0.55 N, high—1.0–1.6 N. The results of length change measurements were recorded using the dedicated software SYLCOM Lite (version 1.5.11.5417) with a sampling time of 0.8 s.

### 3.2. Material

ADI cast iron from an industrial melt made under the conditions of Odlewnie Polskie S.A. foundry, Starachowice, Poland, was used as the material for this study.

The cast iron was melted in an induction furnace with a crucible capacity of 7 t. Alloying elements (Cu and Ni) were added into the furnace together with a charge consisting of pig iron (40%), scrap steel (20%) and scrap circulating iron ADI (40%). The liquid alloy from the furnace was taken into a slender ladle with a capacity of 1 t. In this ladle, by the Thundish Cover method, spheroidization was performed by ferrosilicium with 6.5 wt.% of Mg. After spheroidization, the cast iron was transferred into a pouring ladle and modified with 4 kg of Ba modifier (10 wt.% Ba) and 2 kg of ferrovanadium (80 wt.% V) with graininess of 2–6 mm. The secondary modification was carried out directly on the cast iron stream poured into the mold with ULTRASEED^®^ Ce Innoculant, Elkem AS, Oslo, Norway [42] in an amount of about 0.15 wt.%.

The samples for the tests were taken from the Y 25 type of sample ingots, according to EN 1564 [43], made in the form of a commercial casting.

Plate samples for chemical composition testing, approximately 4.5 mm thick, were cast into a copper die during mold pouring. The chemical composition study was performed on an AMETEK SPECTROLAB spectrometer (AMETEK, Inc., Berwyn, PA, USA). The results of measuring the chemical composition of the resulting cast iron are shown in Table 1.

### 3.3. Heat Treatment and Dilatometric Analysis

The isothermal quenching process consisted of the following procedures. The samples were heated at a rate of 4 K/s from ambient temperature to an austenitizing temperature of 875 °C. The holding time at the austenitizing temperature was 120 min. The nominal temperature of isothermal quenching in the salt bath was 387 °C. The course of changes in the temperature and length of two samples as well as the temperature of the salt bath during the complete heat treatment is shown in Figure 2. The data in this figure are shown on a time scale, with the beginning being when cooling starts. For each sample, the incubation period prior to the start of ausferrite growth (*T*_in_) was determined based on the time to achieve the minimum length during austempering (*L*_min_).

In the following graphs, the zero point of the time scale is moved to the time instance when the nucleation incubation period ends. Figure 3 shows the changes in the temperature of the samples over a time interval from the start of cooling (the time scale is logarithmic). The figure also presents the relative changes in specimen length with respect to their initial length, measured at ambient temperature prior to heating.

The minimum sample length was recorded approximately 50 s after the start of rapid cooling. These moments are marked in Figure 3. They were adopted in the analysis of the measurement results as the start time of austenite decomposition.

## 4. Processing of Experimental Results

### 4.1. Dilatation Analysis

Based on the results of measurements of changes in the length of the samples, changes in the degree of transformation completion *f*_D_ estimated on the base of dilatometry were determined using the following equation:(16)$$f_{D}\left(t\right)=\frac{L\left(t\right)-L_{\min}}{L_{\max}-L_{\min}},$$
where *L*(*t*) is the current value of the measurements of length change and *L*_min_ and *L*_max_, respectively, are the minimum values after the start of austempering and the maximum values before the end of austempering. The results of the measurements in Avrami coordinates (13) are shown in Figure 4.

It has been established that, during isothermal quenching, the kinetics of phase transformations, as evaluated from dimensional changes in the samples, cannot be described by the Avrami Equation (11) with one fixed set of parameters *n* and *k*. Nevertheless, for each of the samples analyzed, it was possible to distinguish four steps of transformation. Each of these steps can be approximated by Equation (11) with different constant parameter values. The values of the linear regression coefficients were determined using the least squares method by solving a system of Gauss–Markov equations [44]. Appendix A provides a detailed description of the mathematical apparatus used to process the measurement results. The results obtained, along with their statistical evaluation performed using the Student’s t-criterion for a confidence level of α = 0.05, are summarized in Table 2.

As can be seen from the data presented, the different steps of transformation in the analyzed samples have similar durations. They are characterized by a high level of the coefficients of determination *R*^2^ and similar values of the coefficient *k* and exponent *n* in the Avrami equation. Variations in the parameters of the Avrami equation may indicate changes in the transformation mechanism [38].

A value of *n* significantly lower than the theoretical value for diffusion-controlled growth (1^1^/_2_) may indicate that ferrite grain growth is additionally controlled by other factors. Since the density of the forming grains is greater than that of the disappearing matrix, this factor for the analyzed samples may be the relaxation rate of the mechanical stresses.

Ferrite grains formed from supersaturated austenite during its decomposition exhibit an anisotropic shape, and their spatial orientations vary. Therefore, another possible reason for the observed changes in the kinetics of the process may be the screening of some surfaces of the growing particles in the case of anisotropic growth of grains with random spatial orientation, as described in [45]. If there are areas on the surfaces of such grains, with different levels of growth speed, the earliest growth will be blocked in the area with the highest speed. That results in a reduction in the exponent *n*. The transition from step A to step B may be related to blocking the growth of platelet faces. The acceleration observed at the transition from step C to step D may signify the crossing of the processing window of austenite decomposition and the start of carbide precipitation (the second stage of austempering). Verifying this is beyond the scope of this article.

The time instants of the completion of steps A, B and C and transition to the corresponding next step for the analyzed samples as well as the levels of austenite decomposition at these moments, calculated according to Equation (16), are presented in Table 3.

### 4.2. Metallografic Analysis

#### 4.2.1. Metallographic Sample Preparation

Metallographic samples for the purpose of analyzing the microstructure of the metal matrix of cast iron were cut with a continuous flow of coolant on a Struers Labotom-5 machine. For automatic grinding and polishing, the samples were mounted in PolyFast resin clamps using the Struers CitoPress-5 hot compression method. The mounted samples were subjected to a sequential grinding process using a Struers Tegramin-20 automatic grinding and polishing machine. For this purpose, MD-Piano discs with the following grain sizes were used: 120, 220, 600, 1200 and 2000. The final surface preparation was achieved by polishing on an MD-Nap disc saturated with a colloidal silica suspension with a grain size of 1 μm. The polished samples were etched in 3% Nital. Etching was conducted by short immersion, followed by immediate rinsing with ethanol and drying with warm air.

The obtained microstructure of the probe for different ausferritization times is presented in Figure 5, Figure 6 and Figure 7, with a magnification 100× to 1000×.

As can be seen from Figure 5, the microstructure of the samples analyzed is homogeneous. The distribution of graphite spheroids is uniform. Figure 6 confirms that, with an increase in isothermal treatment time, both the number of ferrite grains and their volume fraction increase.

At a higher magnification (Figure 7), it becomes apparent that disturbances develop and grow in the flat boundary of the “austenite → ferrite” border as the degree of austenite decomposition increases.

Methods of etching metallographic specimens are used to identify the components of the ADI metal matrix microstructure, allowing ferrite grains to be detected and distinguished from high-carbon austenite. However, the heterogeneity of color and intensity of reflected light hinder the quantitative analysis using computerized image analysis systems. This paper employs stereological measurement methods without computer assistance.

#### 4.2.2. Volume Fraction of the Microstructure Components

The volume fractions of the microstructure components were measured using the point method. For each sample, 25 microphotographs with a resolution of 2880 × 2160 pixels taken at 2000× magnification were analyzed. The length of the pixel side corresponds to an actual dimension of 0.05155 μm on the sample. A measurement grid consisting of 100 measurement points was applied to each photo, so the volume shares were calculated on a set of 2500 points. A view of microphotographs with measuring grids for sample microphotographs of the analyzed samples is shown in Figure 8. The measurement results are summarized in Table 4, Table 5, Table 6 and Table 7. The volume fractions (*f*_x_) of individual phases (x) were determined using the following relationships:(17)$$f_{x}=\frac{N_{x}}{N_{T}}$$where *N*_x_ denotes the number of points per grain of the x-phase and *N*_T_ denotes the total number of measurement points.

A comparison of the data in Table 6 with the results in Figure 4 indicates that the first slowdown in the transformation rate occurs when the volume fraction of the ferrite grains exceeds approximately 50 vol. %.

The result of a basic measurement using the point method can be 1 if the measurement point is located within the field of the analyzed component or 0 if not. Binomial statistical distribution is the most accurate model to analyze the results of volume fractions of a component of the microstructure. Equation (17) defines the estimation of the mean value of such a statistical distribution. The solution proposed by Wilson [46] can be used to determine the confidence interval limits for the mean value of this statistical distribution:(18)$$p_{l/u}=\frac{\hat{p}+\frac{z_{1-\alpha /2}^{2}}{2N_{T}}\pm z_{1-\alpha /2}\sqrt{\frac{\hat{p}\left(1-\hat{p}\right)}{N_{T}}+\frac{z_{1-\alpha /2}^{2}}{4N_{T}^{2}}}}{1+\frac{z_{1-\alpha /2}^{2}}{N_{T}}},$$
where *p*_l/u_ denotes the lower and upper limits of the confidence interval, respectively, α is the significance level, and *z*_1−α/2_ is the quantile of the standardized normal distribution for probability equal to 1 − α/2 (so-called *z*-function).

The measurement results and their statistical evaluation are summarized in Table 4, Table 5, Table 6 and Table 7. The obtained results show the changes in the mean volume fractions of microstructure components, with limits of the confidence intervals. The limits were calculated for the significance level of α = 0.05 (95% confidence level). These results are also presented in Figure 9.

Figure 10 illustrates the relationship between the experimentally determined volume fraction of ferrite and the results of the degree of transformation completion assessment based on dilatometric measurements. As can be seen from this figure, in the observed time range of austempering, the relationships between these variables and the logarithm of the heat treatment time are close to linear. The values of the determination coefficients are close to 1 for both relationships.

The regression equations shown in Figure 10 suggest a direct linear relationship between the volume fraction of ferrite *f*_F_ in ausferrite and the degree assessment of transformation completion by the dilatometric method in the analyzed time interval:(19)$$f_{F}=-0.0669+0.783\cdot f_{D}.$$

The answer to the question of whether similar relationships will occur in other ADI cast iron grades is beyond the scope of this article. This topic will be the subject of further research.

#### 4.2.3. Practical Significance of the Results Obtained

The precipitation of carbide particles from high-carbon austenite results in a decrease in density and an increase in alloy volume. This is why the transition from step C to step D, as indicated by an increase in slope on the Avrami diagram (Figure 4), may also indicate the beginning of the second stage of the isothermal austenite decomposition.

As shown in Figure 4 and Table 2, the slope angle of the Avrami diagram in step C differs significantly from the course of this diagram in adjacent sections. The increase in elongation during this stage of transformation is significantly less than 1%. As indicated by relation (19), the change in the volume fraction of ferrite should be at a similar level. In both analyzed samples, step C begins after approx. 1 h 26 min (~5200 s) from the moment of incubation and lasts over an hour (115 min in sample No. 1 and 73 min in sample No. 2). This means that continuing the treatment at stage C will not be effective, at least from an economic point of view.

The shape of the diagrams obtained, shown in Figure 4, indicates that the critical value of the angular slope coefficient of the Avrami graph can be taken as the criterion for determining the moment of transition from form B to C:(20)$$K_{C}=\frac{d}{d\ln\left(t\right)}\left\{\ln\left[-\ln\left(1-f_{D}\right)\right]\right\}.$$

As can be seen in Table 4, for cast iron with the analyzed chemical composition and heat treatment parameters, it can be assumed that this value is *K*_C_ ≈ 0.5.

#### 4.2.4. Specific Area of the Boundary Surface

The areas of the grain boundary surface per unit volume for grains of the different phases were evaluated using the point method. For the same analytical grids used for the volume fraction measurements, the number of points *N*’ where grid lines intersect the grain boundaries was calculated for each type of contact. The specific area for each type of grain boundary *s*_x-y_ was calculated by dividing the obtained point number *N*_x-y_ by the total length of the analytical grid line L:(21)$$s_{x-y}=\frac{N_{x-y}^{\prime }}{L},$$
where x and y are the phases neighboring this boundary. The unit of this value is [m^−1^], which is equal to [m^2^/m^3^].

The measurements were performed based on five microphotographs for each probe. The common length of the grid lines for one probe *L* was equal to 12.99 mm.

The measurements identified the boundaries between ferrite grains and austenite, graphite and other ferrite grains. The austenite–graphite boundaries were also analyzed. Since the austempering temperature was too low for austenite recrystallization to occur, no etching was used to detect the austenite grain structure. The primary results of the measurements are presented in Table 8.

As can be seen from this table, there is a tendency for the contact area between the graphite spheroids and the ferrite grains to increase and for the contact area between the graphite and the austenite to decrease. As austenite decomposes, the contact area between the disappearing austenite and the growing ferrite grains increases.

An increase in austempering time results in a greater difference in the full surface areas of ferrite and the contact areas with austenite, where ferrite growth continues (Figure 11). This difference indicates the density of the internal grain boundaries between the ferritic grains.

The *N*_X-Y_ values listed in Table 8 represent the average numbers if intersection points between the boundaries of grains of the corresponding type and the 12.99 mm measuring lines are used. The statistical distribution of this variable is consistent with the Poisson distribution. For this distribution with the expected value *N*, the probability of obtaining a specific value of *k* in the same conditions is given by *p*(*k*)(22)$$p\left(k\right)=\frac{e^{-N}N^{k}}{k!}.$$

The variance of the Poisson distribution is equal to its mean value, and the standard deviation σ is equal to the square root of this value. For mean values above 200, the Poisson statistical distribution is very close to the normal distribution, with corresponding mean and standard deviation values. Therefore, in Figure 11, in accordance with the “six sigma” principle, the height of the error bars for the measurement data is equal to six times the standard deviations (±3·σ).

#### 4.2.5. Assessment of Ferrite Grain Shape

To evaluate changes in the thickness and shape of ferrite grains, the *M* parameter was used, calculated as the result of dividing the volume of ferrite grains by the contact area of these grains with the disappearing austenite. The specific values of these parameters were used to calculate this assessment:(23)$$M=\frac{f_{F}}{s_{F-A}}.$$

This is the equivalent of the geometric module used in the foundry industry to evaluate the geometry of local sections of a casting. The ratio of volume to surface area of a body is a quantity measured in units of length (e.g., meters). This rating shows how much of this body’s volume is per unit of its surface area. The results of this assessment are shown in Figure 12.

The results obtained indicate that, during step A (samples a, b and c; Figure 4 and Table 2), as the isothermal treatment time increases, the geometric modulus of the ferrite grains increases. This is typical for the initial phase of growth of grains without mutual impingements during phase transformations in the solid state or crystallization. The significant reduction in geometric modulus observed at step B (probe d) is associated with the blocking of rapid growth on the frontal surfaces of needle-like grains. The resulting “blocking” effect, described in Chapter 5, causes a reduction in the slope of the Avrami diagram.

An additional source of reduction in the geometric modulus of ferrite grains is branching on the slowly growing side surfaces of ferrite grains. As a result, a wheat-ear-like grain structure begins to form. A comparison of the microstructure of samples a–d shown in Figure 7 shows that grains of this type appear in greater numbers only in sample d.

### 4.3. Future Research Directions

Dilatometric analysis was performed on two samples made of cast iron obtained from the same casting. Both samples exhibited similar changes in the kinetics of austenite decomposition, namely three steps of austenite decomposition, which differ in terms of the kinetics of the transformation (sections A, B and C in Figure 4). Based on the results of the metallographic analysis, the mechanism of phase transformation slowdown causing the transition from step A to B was explained.

Considering the results obtained, one direction for further research should be to confirm the hypothesis that the transition from step C to step D is associated with the initiation of the second stage of austenite decomposition. In this case, the acceleration of dimensional changes may be related to the release of carbides, i.e., phases with lower density. Of course, testing this hypothesis requires using direct verification by research methods of microstructure analysis.

Further research should be conducted to verify whether similar changes in the kinetics of isothermal decomposition of austenite occur in cast iron with a different chemical composition and under different isothermal quenching conditions (austenitization and ausferritization temperature and time). Additionally, further investigation is needed to ascertain whether the value of the proposed *K*_C_ criterion calculated by Equation (20) varies with changes in the chemical composition of the alloy or the heat treatment conditions.

The mechanism of slowing down the transformation during the transition from step B to C also remains to be investigated.

## 5. Conclusions

A dilatometric analysis was performed on changes occurring during isothermal austempering of ADI cast iron samples with the chemical composition presented in Table 1. The results were compared with a qualitative and quantitative analysis of the microstructure of four samples frozen during exothermic heat treatment at 387 °C after austenitization at 875 °C.

It has been shown that there is a linear relationship between the assessment of the degree of austenite decomposition using the dilatometric method and the volume fraction of ferrite grains during the initial phase of the first stage of austenite decomposition. For the analyzed alloy and its heat treatment conditions, this relationship is expressed by Equation (22).

To analyze the kinetics of austenite decomposition, the results of dilatometric measurements were processed using the Avrami transformation—Equation (11). It was established that, during the first stage of austenite decomposition, three sections can be separated on the Avrami diagram, within which the kinetics of the transformation are described by equations of type (11) with a different set of parameters. These changes are statistically significant and occur twice (Table 2). The observed changes indicate the disappearance of faster migration processes at the ferrite–austenite phase boundary.

It has been shown that the first of the observed slowdowns in transformation occurs when the volume fraction of ferrite exceeds 50%. This is associated with the blocking of areas of rapid growth at the front of needle-like ferrite grains by the slowly migrating side walls of other grains of this type. This causes further transformation to take place through slower movement of the side walls of these grains. At the same time, the shape of these side surfaces of ferrite grains begins to change from smooth to wheat-ear-like.

## Figures and Tables

**Figure 1 materials-18-05039-f001:**
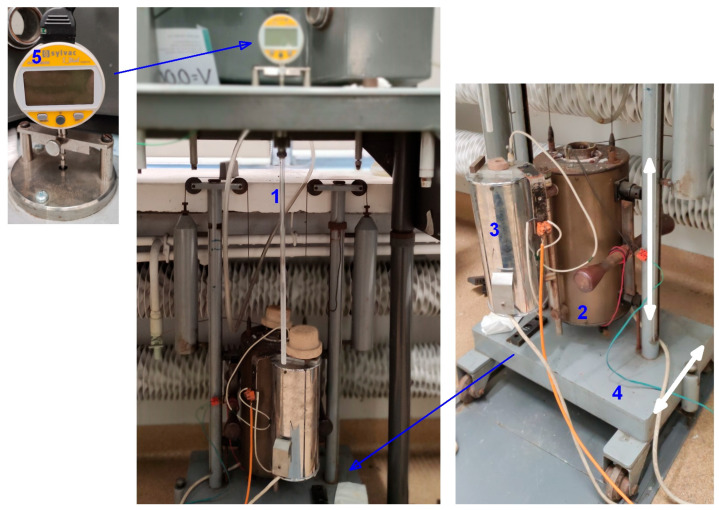
Laboratory equipment for isothermal quenching: (1) quartz ampoule with sample; (2) high-temperature austenitizing furnace; (3) salt bath for isothermal quenching; (4) platform for moving furnaces in the directions indicated by the arrows; (5) elongation measurement device; white arrows indicate the directions of movement of the installation elements.

**Figure 2 materials-18-05039-f002:**
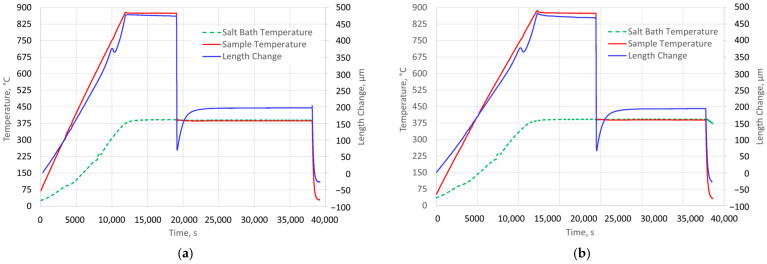
The course of temperature changes in the samples and the salt bath as well as changes in the length of the samples throughout the heat treatment: (**a**) for sample No. 1; (**b**) for sample No. 2.

**Figure 3 materials-18-05039-f003:**
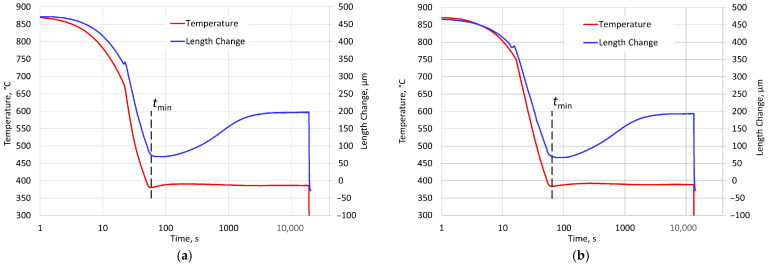
The course of temperature changes (red lines) in the specimens and their length changes (blue lines) from the moment of the beginning of quenching: (**a**) for specimen No. 1; (**b**) for specimen No. 2; *t*_min_—the moment of reaching the minimum length of the specimen.

**Figure 4 materials-18-05039-f004:**
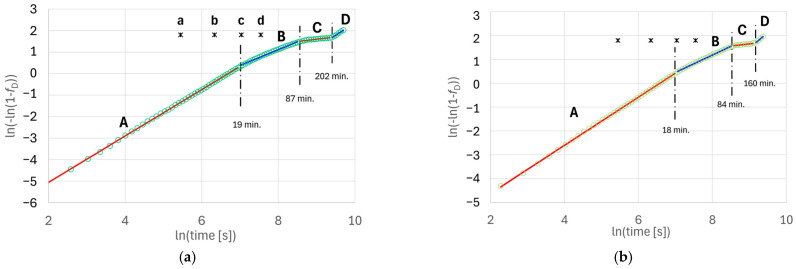
Changes in the degree of phase transformations in Avrami diagrams: (**a**) sample no. 1, (**b**) sample no. 2; circles—experiment; lines—regression (different colors of the lines indicate the different steps of transformation A–D, as shown in Table 2); points a–d—time instants of the samples quenching for microstructure analysis.

**Figure 5 materials-18-05039-f005:**
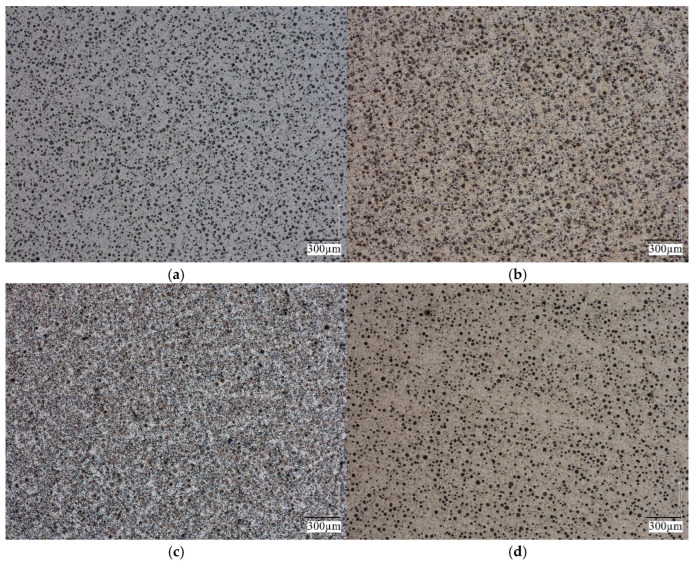
Microstructure of iron for austempering time (**a**) 232 s, (**b**) 564 s, (**c**) 1134 s and (**d**) 1891 s; magnification 100×.

**Figure 6 materials-18-05039-f006:**
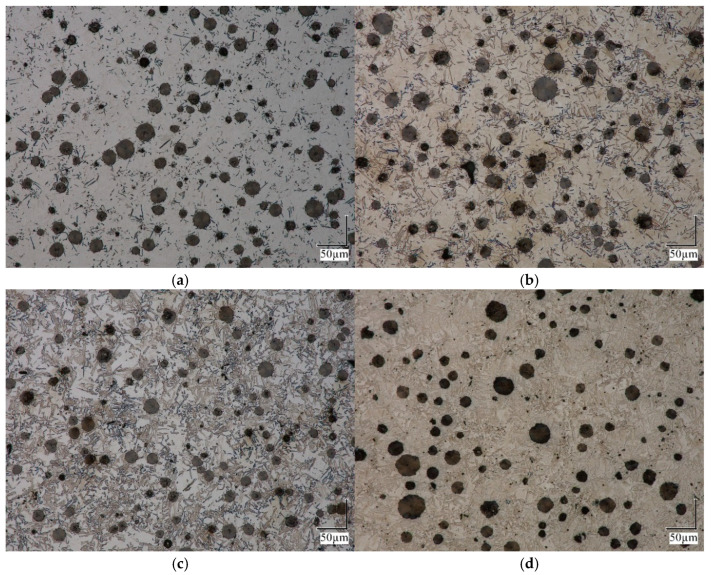
Microstructure of iron for austempering time (**a**) 232 s, (**b**) 564 s, (**c**) 1134 s and (**d**) 1891 s; magnification 500×.

**Figure 7 materials-18-05039-f007:**
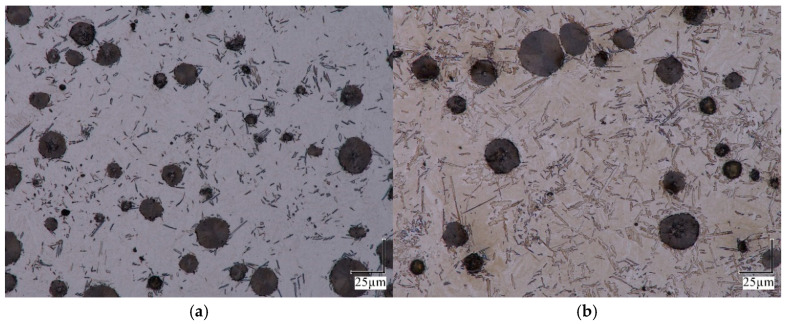
Microstructure of iron for austempering time (**a**) 232 s, (**b**) 564 s, (**c**) 1134 s and (**d**) 1891 s; magnification 1000×.

**Figure 8 materials-18-05039-f008:**
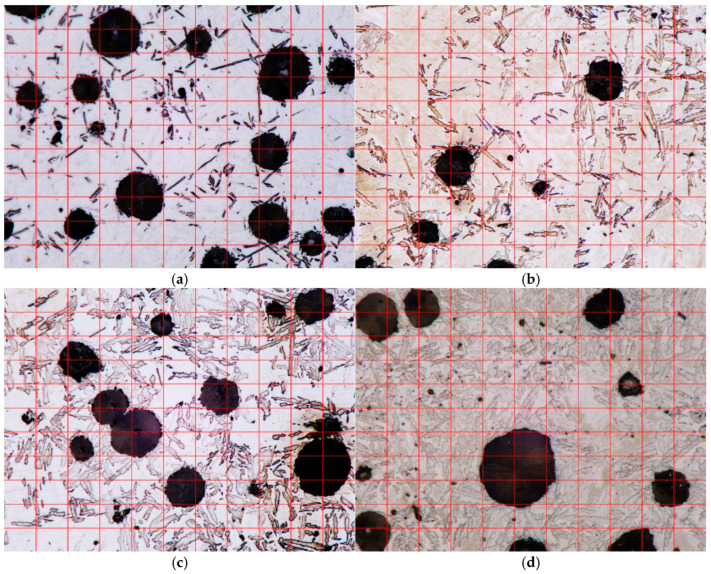
Measuring grids (red lines) on microphotographs taken at 2000× magnification for austempering time (**a**) 232 s; (**b**) 564 s; (**c**) 1134 s and (**d**) 1891 s.

**Figure 9 materials-18-05039-f009:**
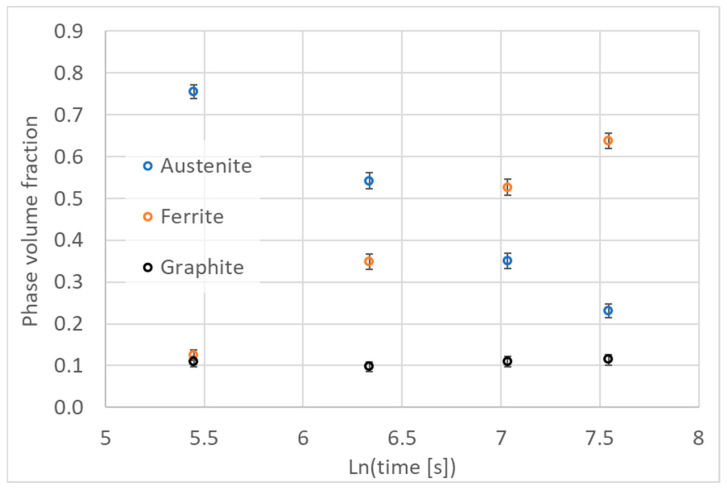
Changes in volume fraction of microstructure components during austempering treatment; height of the error bars is according to Equation (18) and Table 4, Table 5, Table 6 and Table 7.

**Figure 10 materials-18-05039-f010:**
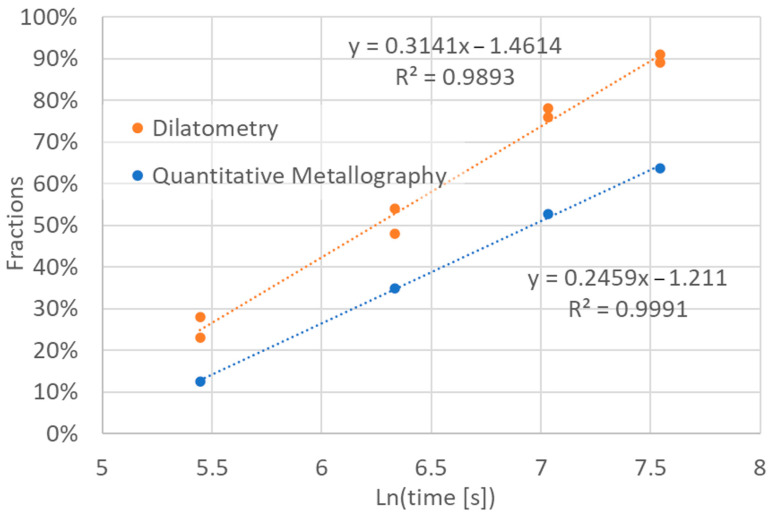
Changes in degree of transformation completion (dilatometry) and of ferrite volume fraction (quantitative metallography).

**Figure 11 materials-18-05039-f011:**
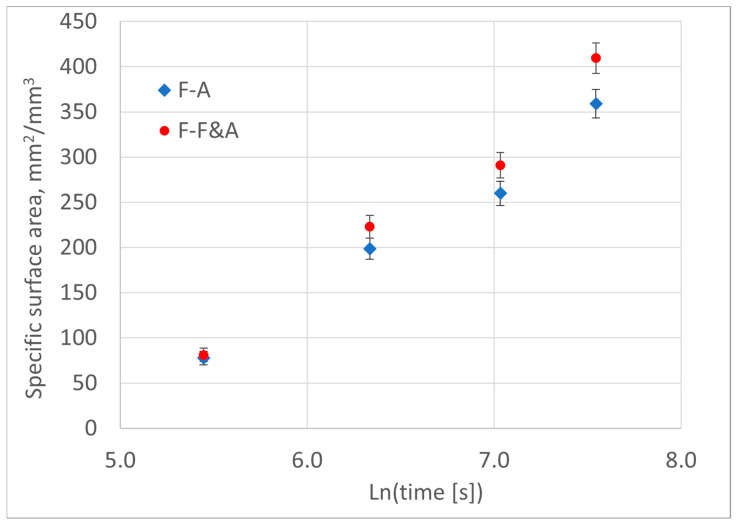
Specific surface area of outer boundaries of ferrite grains (F-A) and all boundaries of ferrite grains (F-F&A) within the cast iron metal matrix.

**Figure 12 materials-18-05039-f012:**
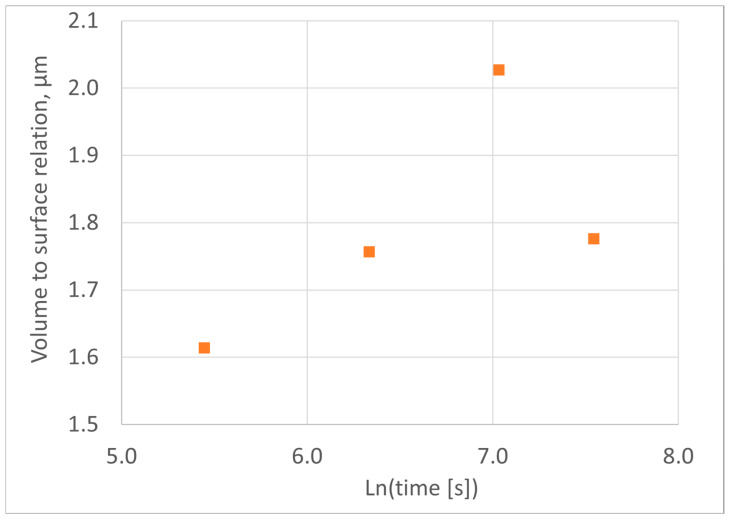
Assessment of volume to surface relation for ferritic grains.

**Table 1 materials-18-05039-t001:** Chemical composition of ADI, wt.% *.

**C**	**Si**	**Mn**	**Mg**	**Cu**	**Ni**	**Mo**	**Cr**
3.43	2.33	0.11	0.048	0.72	2.64	0.01	0.04
**V**	**Nb**	**Zn**	**Co**	**Se**	**Al**	**Ca**	**P**
0.0689	0.0147	0.0619	0.0251	0.0156	0.0144	0.0109	0.029

* Content of S, Zr, Ce, La, Bi, B, As, Sb, Su, W, Ti, Nb and Pb does not exceed 0.01 wt.%. The total content of these components is 0.0565 wt.%.

**Table 2 materials-18-05039-t002:** Results of statistical evaluation of linear regression parameters (11).

Step	Sample	Start Time, min	Exponent *n*			Coefficient *(k)*			*R* ^2^
			Mean Value	Lower Limit	Upper Limit	Mean ln(*k*)	*k* Lower	*k* Upper	
A	1	0	1.077	1.044	1.110	−7.209	6.18 × 10^−4^	7.65 × 10^−4^	0.9998
	2	0	1.009	0.978	1.040	−6.634	1.11 × 10^−3^	1.36 × 10^−3^	0.9998
B	1	19	0.740	0.628	0.852	−4.807	3.42 × 10^−3^	9.1 × 10^−3^	0.9947
	2	18	0.730	0.662	0.799	−4.658	5.55 × 10^−3^	1.02 × 10^−2^	0.9980
C	1	87	0.211	0.087	0.335	−0.298	2.44 × 10^−1^	8.4 × 10^−1^	0.9319
	2	84	0.178	0.070	0.286	0.048	4.04 × 10^−1^	1.168	0.9384
D	1	202	1.162	0.983	1.341	−9.253	1.73 × 10^−5^	1.15 × 10^−4^	0.9994
	2	160	1.294	1.141	1.446	−10.165	9.38 × 10^−6^	4.48 × 10^−5^	0.9993

**Table 3 materials-18-05039-t003:** Completion time of steps A, B and C and levels of transformation completion (according to Avrami diagram).

Step	Completion Time, s		Assessment of Transformation Completion Degree, %	
	Sample 1	Sample 2	Sample 1	Sample 2
A	1114	1081	75.1	77.5
B	5239	5213	98.8	99.1
C	12,124	9612	99.5	99.6

**Table 4 materials-18-05039-t004:** Statistical evaluation of volume fractions for ausferritization time 232 s.

Phases	Quantity of Points	Mean Volume Fraction	Confidence Interval Limits	
			Lower	Upper
Austenite	1886	75.4%	73.7%	77.1%
Ferrite	313	12.5%	11.3%	13.9%
Graphite	275	11.0%	9.8%	12.3%
Non-identified	26	1.0%	0.7%	1.5%
Total	2500			

**Table 5 materials-18-05039-t005:** Statistical evaluation of volume fractions for ausferritization time 564 s.

Phases	Quantity of Points	Mean Volume Fraction	Confidence Interval Limits	
			Lower	Upper
Austenite	1354	54.2%	52.2%	56.1%
Ferrite	872	34.9%	33.0%	36.8%
Graphite	246	9.8%	8.7%	11.1%
Non-identified	28	1.1%	0.8%	1.6%
Total	2500			

**Table 6 materials-18-05039-t006:** Statistical evaluation of volume fractions for ausferritization time 1134 s.

Phases	Quantity of Points	Mean Volume Fraction	Confidence Interval Limits	
			Lower	Upper
Austenite	877	35.1%	33.2%	37.0%
Ferrite	1317	52.7%	50.7%	54.6%
Graphite	276	11.0%	9.9%	12.3%
Non-identified	30	1.2%	0.8%	1.7%
Total	2500			

**Table 7 materials-18-05039-t007:** Statistical evaluation of volume fractions for ausferritization time 1891 s.

Phases	Quantity of Points	Mean Volume Fraction	Confidence Interval Limits	
			Lower	Upper
Austenite	578	23.1%	21.5%	24.8%
Ferrite	1593	63.7%	61.8%	65.6%
Graphite	286	11.4%	10.3%	12.7%
Non-identified	43	1.7%	1.3%	2.3%
Total	2500			

**Table 8 materials-18-05039-t008:** Primary results of grain boundary density measurements.

Sample	Number of Points at Borders				
	*N* _F-A_	*N* _F-F_	*N* _F-G_	*N* _A-G_	Total
a	1008	47	35	142	1185
b	2580	316	103	68	2751
c	3377	403	166	81	3624
d	4662	657	170	27	4859

Note: Subscripts indicate phases occurring on both sides of the border: A—austenite, F—ferrite, G—graphite.

## Data Availability

The original contributions presented in this study are included in the article. Further inquiries can be directed to the corresponding authors.

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
