# Peer review of "Application of the Avrami Equation to the Dilatometric Analysis of ADI Austempering Kinetics"

_materials, 2025, doi:10.3390/ma18215039_

Round 1

Reviewer 1 Report

Comments and Suggestions for Authors

This paper investigated the multi-step kinetic characteristics of the first stage of austenite decomposition in ADI through the combination of dilatometric analysis and the Avrami equation. The research results is very interesting and useful to the guide the industrial application. The specific comments are as follows.

  1. In the introduction, the place for the experiment should be deleted.
  2. This paper focuses solely on austempered ductile iron (ADI) samples with a specific chemical composition (0.72% Cu and 2.64% Ni) and investigates a single isothermal temperature of 387°C. It does not address the impact of key parameters such as the content of alloying elements (e.g., Mo, Mn), austenitizing time, and isothermal temperature on the fitting parameters (n and k values) of the Avrami equation and the kinetics of phase transformation. To enhance the comprehensiveness of the study, it is recommended to conduct additional comparative experiments using varying chemical compositions and isothermal temperatures to analyze how these variables influence the characteristics and rates of phase transformation stages.
  3. In Table 2, the statistical calculation basis for "n lower" and "n upper," particularly the method for calculating confidence intervals, is not adequately detailed, potentially undermining the credibility of the data. It is recommended to include the calculation method for the upper and lower limits of the n value, such as the derivation process for confidence intervals based on Student's t-test, in the notes accompanying Table 2. Additionally, a brief explanation of this methodology should be provided in the "4.1 Dilation Analysis" section of the main text to enhance the transparency of the data.
  4. The current Section 2 fails to provide a systematic review of the application of the Avrami equation in the analysis of phase transformations in austempered ductile iron (ADI). It is recommended that a comparative analysis of relevant literature be supplemented alongside the methods utilized in this paper. Additionally, the "1. Introduction" section should articulate the role of this study in addressing deficiencies in existing methodologies while emphasizing the research's innovative contributions.

Reviewer 2 Report

Comments and Suggestions for Authors

This work focused on the application of the avrami equation to the dilatometric analysis of ADI austempering kinetics. The topic is meaningful and some good results were obtained. However, the authors should address the following issues before this paper is accepted for publication in Materials.

(1) In 3.2 section, the authors pointed out that “a crucible capacity of 7 Mg” and “a capacity of 1 Mg”, what does “Mg” mean?

(2) In Fig.4, what are the phase transition mechanisms corresponding to stages A, B, C, and D? Interface control or diffusion control? Can the growth activation energy of each stage be determined through dynamic model analysis?

Reviewer 3 Report

Comments and Suggestions for Authors

This paper examines the kinetics of isothermal decomposition of austenite in high-strength ductile iron (ADI) using dilatometric analysis and the Avrami equation. The study demonstrates the feasibility of using the Kolmogorov–Johnson–Mehl–Avrami theoretical model to analyze dilatometry data, enabling the identification of the stages of ausferrite structure formation and statistical evaluation of changes in the phase transformation mechanism. The authors demonstrated the existence of three sequential stages of austenite decomposition and demonstrated that the slowdown in the transformation rate is associated with the screening of the growth of acicular ferrite grains. The obtained relationships between the ferrite fraction and dilatometric parameters provide the basis for more accurate prediction of the process "window" for heat treatment in industrial settings. This paper is relevant for foundries, as it enables optimization of ADI heat treatment parameters and improvement of the homogeneity and performance properties of castings. The study combines a fundamental approach with practical applications in materials science and engineering. However, the article needs some revision:
1. At the end of the review of reasons/methods, you formulate the goal as "analyzing changes in the kinetics of the first stage and determining its completion point." It would be useful to immediately describe the engineering motivation: how this will shorten the process window selection cycle in the shop (e.g., switching from a series of quenches to online dilatometry) and what metrics (time error, sample savings) you consider a practical success. This would strengthen the practical significance. (lines 243–246).
2. Section 2.1 qualitatively summarizes the approaches but does not compare them with your protocol (cooling rates, pearlite elimination method, TCLE adjustments, t0 selection). Add a summary table: method/sensor/resolution/axial load/speed/start/finish metric. This will allow the reader to understand where exactly your methodology provides an increase in information content compared to Santos, Gazda, Ahmed, and others (lines 92–121, 106–115).
3. In 2.2, equations (3)–(7) are presented correctly, but it remains unclear why you choose the linearized Avrami form for interpretation, rather than, for example, AR with free fmax, given the incomplete transformation in the given window. Reasoning is needed (approximation quality, interpretability of n, robustness to length-measuring noise, sensitivity to the choice of Lmax/Lmin).
4. The alloy composition and modification route are described in detail, which is valuable for reproducibility. However, Table 1 contains a note about "total content of other... 2.215 wt.%," which appears to be a typo (probably 0.2215% or similar). Rechecking units and amounts is required.
5. Errors and distributions. You refer to Poisson and "six sigma"; the caption to Fig. 9 says "3 SD," while the text says "6 SD"; there is an inconsistency that requires consistency. Furthermore, for fractions close to 0/1, the normal approximation is inaccurate; Wilson's binomial confidence intervals for phase fractions can be used. (lines 431–439; 434–436).
6. Interpretation of section D. You cautiously write that the acceleration may be related to the onset of stage II and the precipitation of carbides. It is important to separate this from the proven facts. Currently, direct verification is lacking (no XRD/Temp-dependent hardness/DSS). Propose a verification plan in "Future work" and soften the wording in "Conclusions." (lines 521–524; 532–548).
7. For Avrami linear plots, it is useful to add R², RMSE, and residual plots, as well as to show the robustness of the slopes to variations in t₀ (the zero shift in Tin). Small shifts in t₀ can significantly change n.
8. The stated goal is to determine the completion point of stage I. According to Table 3, you provide completion times for A, B, and C and levels of fD≈99–99.6%. It would be good to explicitly state the "completion" criterion (e.g., dfD/dt below the threshold ε for Δt) and demonstrate its applicability to industrial timings (in minutes/hours), so that the reader can immediately transfer it to process charts. (lines 363–367).

Round 2

Reviewer 3 Report

Comments and Suggestions for Authors

Important methodological and statistical concerns (3, 5, 6) are addressed thoroughly and effectively. This significantly strengthens the article. Practical motivation and connections to production are convincingly added (1). In terms of formatting and data, typos have been corrected (4), and terminology has been streamlined (9). One area remains a "half-step" (7): RMSE, residuals, and the demonstration of robustness of n to t₀ were not added to the main text (the t₀ method is described only in the letter).

Author Response

Dear Reviewer,
We are very grateful for all the questions and comments provided in the review of the first version of our manuscript. We agree that the revisions made in response to these remarks have significantly contributed to improving the quality of the publication. We also believe that the reviewer’s suggestions will be valuable in the preparation of future work continuing this research topic.
In preparation for the next publication, we are revising the experimental methodology, measurement procedures, and statistical analysis. In that subsequent work, we plan to present the second “half-step” that was not included in the current manuscript.
Thank you once again for your constructive feedback.
Sincerely,
Tomasz Wiktor